# Multi-Winner Election Control via Social Influence: Hardness and Algorithms for Restricted Cases

**Mohammad Abouei Mehrizi \***  **and Gianlorenzo D'Angelo \***

Computer Science Department, Gran Sasso Science Institute (GSSI), Viale Francesco Crispi, 67100 L'Aquila AQ, Italy

\* Correspondence: mohammad.aboueimehrizi@gssi.it (M.A.M.); gianlorenzo.dangelo@gssi.it (G.D.)

**Abstract:** Nowadays, many political campaigns are using social influence in order to convince voters to support/oppose a specific candidate/party. In election control via social influence problem, an attacker tries to find a set of limited influencers to start disseminating a political message in a social network of voters. A voter will change his opinion when he receives and accepts the message. In constructive case, the goal is to maximize the number of votes/winners of a target candidate/party, while in destructive case, the attacker tries to minimize them. Recent works considered the problem in different models and presented some hardness and approximation results. In this work, we consider multi-winner election control through social influence on different graph structures and diffusion models, and our goal is to maximize/minimize the number of winners in our target party. We show that the problem is hard to approximate when voters' connections form a graph, and the diffusion model is the linear threshold model. We also prove the same result considering an arborescence under independent cascade model. Moreover, we present a dynamic programming algorithm for the cases that the voting system is a variation of straight-party voting, and voters form a tree.

**Keywords:** computational social choice; election control; multi-winner election; social influence; influence maximization

## 1. Introduction

Social media is an integral part of nowadays life. No one can ignore the effect of social media on different aspects of our life. Many people from all around the world are using social networks to provide/use various services like teaching/learning, spreading information, events' announcements, and advertising. It has been shown that two-thirds of American adults get news on social mediaSM [1]. It is easy to find evidence that a social influence (SI) started by few users has influenced many people. Then, social media is a kind of cheap means to spread a message among many users. Note that the power of social media is not just like spreading a message or advertising. Its power comes from the fact that a user will receive news from those who have enough authority to change his opinion, like close friends, family members, and colleagues. Since using social influence is effective and cheap, it has been attracting the attention of many political campaigns and candidates to target the user's opinion through SI. They disseminate a piece of information to change voters' opinion. Many real case studies show that campaigns used social influence to change the voters' opinion [2–5]. For example, Allcott and Gentzkow showed that 92% of Americans remembered pro-Trump false news, and 23% remembered pro-Clinton fake news [6].

There are two well-known diffusion models used in social influence called linear threshold model (LTM) and Independent Cascade Model (ICM) [7]. In LTM, a voter accepts a message if the sum over his incoming neighbors' influence, who already accepted the message, is high enough. On the other hand, in ICM, a voter will accept a message if at least one of his incoming neighbors, who already

accepted the message, can convince him to accept it (please see Section 2 for a formal definition of LTM and ICM).

In this paper, we consider the multi-winner election control via social influence problem. We are given a social network of voters, a limited budget, a set of candidates each belongs to a party, a dynamic diffusion model to spread a message among the voters, and an attacker/manipulator who supports/opposes a party. When we use LT diffusion model, we assume that the attacker knows the probability that each voter wants to vote for each candidate. To take into account the incoming influence of each node $v$, we use an updating rule based on the incoming influence from the node's incoming activated neighbors, akin to [8]. On the other hand, when we use ICM, we assume the attacker knows the exact preferences list of all voters. When a node/voter becomes active/influenced/infected, in constructive (resp. destructive) case, it will promote (resp. demote) the position of the target candidates in its/his preference list, akin to [9,10] (see Section 3 for formal definition).

Regarding both LTM and ICM, there will be several winners, and they will be elected according to the overall candidates' scores after the diffusion. In the constructive (resp. destructive) case, the attacker wants to find a set of nodes (voters), according to its budget, to start the diffusion and change the voters' opinion to maximize (resp. minimize) the number of winners from his target party. In fact, in a given directed graph, we should find some diffusion starters to influence the voters such that the difference between the number of winners from our target party, w.r.t. the number of winners in the opponent party with the most winners, after and before the diffusion is maximized (resp. minimized). We present some results, including hardness of approximation, approximation, and polynomial-time exact algorithms considering some well-known objective functions on different structures.

Related works. There are many articles regarding voting manipulation (see the survey in [11]). The problem of finding a set of limited seed nodes from a given graph to maximize the expected number of influenced nodes is known as Influence Maximization (IM) problem. There exists an extensive literature about it, too [12]. Domingos and Richardson [13,14] introduced the IM problem, and Kempe et al. formalized it [7,15]. On the other hand, few works consider both of them together, i.e., the election control through social influence problem.

Wilder and Vorobeychik introduced the election control through SI problem regarding single-winner elections [10]. They investigated maximizing margin of victory (MoV) and probability of victory (PoV), where MoV is the difference of the score between the target candidate and the most voted opponent after and before the diffusion. The problem is considered under ICM. They showed maximizing MoV is *NP*-hard, and presented a $1 - \frac{1}{e}$-approximation algorithm concerning the optimal solution. Furthermore, for maximizing PoV, they showed that it is *NP*-hard to approximate the problem within any constant factor. Corò et al. [16,17] extended the work using any non-increasing scoring function under LTM. They demonstrated the same approximation factor for it. Abouei Mehrizi et al. considered the problem when the attacker knows a probability distribution over the candidates instead of the exact preferences list, under LTM [8]. They showed that maximizing/minimizing the expected probability to vote for a target candidate is hard to approximate within any constant factor under unique game with small set expansion conjecture. They also presented some constant factor approximation algorithms for a relaxed version of the problem. Abouei Mehrizi and D'Angelo showed that in multi-winner elections, when the manipulator wants to maximize/minimize the number of winners in his target party, the problem is inapproximable under ICM, except $P = NP$ [9]. They also presented some constant factor approximation algorithms when the voting system is similar to the straight-party voting.

Bredereck and Elkind considered some different models, like bribing nodes/voters, adding or deleting edges under LTM. They showed that the problem is hard in those models. They also presented some polynomial-time algorithms for specific cases of the problem [18]. Castiglioni et al. investigated similar models under ICM. They showed that the problem is hard even in restricted structures. Regarding the bribing nodes to influence other voters, they proved that the election control is hard even if the given graph is a line. Furthermore, considering the edge removal/addition

case, they demonstrated that the problem is hard even if the attacker has an infinite budget [19]. Faliszewsk et al. considered the problem where each voter has a preference list. Each node of the graph is representative of all users with the same opinions. There is an edge between two nodes if their opinion differs by the place of an adjacent pair of candidates. They used LTM and proved that maximizing the number of votes for the target candidate is *NP*-hard and fixed parameter tractable with respect to the number of candidates [20]. Furthermore, there is another model in which voters have a preference list over candidates, and voters will change their preference list according to the majority of their neighbors' opinions [21–23].

Outline and our results. In Section 2, we define the most prominent diffusion models in the literature (called LTM and ICM) that we used in this paper. Section 3 defines our model and objective functions formally. We show that our problem is hard to approximate within any factor in a general graph when the diffusion model is LTM in Section 4. Section 5 contains the same result when the diffusion model is ICM, and the given graph is in the form of an arborescence, i.e., edges are from leaves to root of the tree. Moreover, in Section 6, we investigate the problem while the voting system is a variation of straight-party voting, where voters can vote for the parties. In other words, voters have a preference list (or probability distribution) over the candidates, but they can vote for the parties instead of candidates. We presented a polynomial-time algorithm based on the dynamic programming approach to find the maximum difference of votes for our target party before and after diffusion. It also gives a $\frac{1}{3}$ and $\frac{1}{2}$-approximation algorithms for maximizing MoV in constructive and destructive models, respectively. Finally, we will discuss the results and future works in Section 7.

## 2. Background

In this section, we introduce two diffusion models that we have used in this paper, called linear threshold model (LTM) and independent cascade model (ICM) presented by Kemp et al. [7,15]. They are the most prominent dynamic diffusion models used in literature (see a survey on the topic [24]).

### 2.1. Linear Threshold Model

We are given a directed graph $G = (V, E)$. Each edge $(u, v) \in E$ has a weight $b_{u,v} \in [0, 1]$. The sum of the incoming weight to each node $v \in V$ is at most one, i.e., $\sum_{u \in N_v^i} b_{u,v} \leqslant 1$, where $N_v^i$ is the set of incoming neighbors of $v$. Furthermore, each node $v \in V$ has a threshold $t_v \in [0, 1]$ which is generated uniformly at random.

In this model, the diffusion will start from a set of nodes $S \subseteq V$ known as seed nodes. At the first step, just the seed nodes will become active/influenced/infected, and all other nodes are inactive. Let us show $A_i$ as the set of nodes that are active at step $i$, i.e., $A_1 = S$. The activation process, for each step $i > 1$, is as follows: all nodes in $A_{i-1}$ will remain active at step $i$, i.e., $A_{i-1} \subseteq A_i$; moreover, each inactive node $v \in V \setminus A_{i-1}$ will become active if the sum of the weight from its incoming activated neighbors is not less than its threshold, i.e., for each node $v \in V \setminus A_{i-1}$, it will be in $A_i$ if $\sum_{u \in N_v^i} b_{u,v} \geqslant t_v$. The diffusion process will proceed in utmost $|V|$ discrete steps, and it will stop as soon as no extra node becomes active, i.e., it stops at step $k > 1$ if $A_k = A_{k-1}$. We use $A_S$ as the set of activated nodes after the diffusion process started from the set of seed nodes $S$. In what follows, to increase the readability of this article, when we say after $S$, it means after the diffusion process started from a set of seed nodes $S$. Note that the thresholds are not a part of the input, and they will be generated uniformly at random and independently when we run the process. Furthermore, the process is random, and several executions on the same graph may get different results for $A_S$.

Kemp et al. [7] defined the IM problem as: Given a graph $G = (V, E)$ and a budget $B \leqslant |V|$. Find a set of seed nodes $S \subseteq V$, ($|S| \leqslant B$) so that the expected $|A_S|$ is maximized. They proved that the problem is *NP*-hard under LTM. Moreover, they showed that a greedy algorithm can solve the problem approximately within a factor of $1 - \frac{1}{e} - \epsilon$, where $\epsilon$ is any small constant and fixed number.

### 2.2. Independent Cascade Model

Consider a graph $G = (V, E)$ with a weight $b_{u,v} \in [0, 1]$ on each edge $(u, v) \in E$. The same as LTM, all nodes are inactive, and at the first step the seed nodes $S \subseteq V$ become active. Let us define $S_i$ as the nodes that were inactive at step $i - 1$ and became active at step $i$, then $S_1 = S$. At each step $i > 1$, each node $v \in S_{i-1}$ will try to activate its outgoing neighbors with the probability of the edge between them. In other words, consider $N_v^o$ as the set of outgoing neighbors of node $v$; for each $u \in N_v^o$, node $v$ tries to activate $u$ with the probability $b_{v,u}$. If $v$ has multiple outgoing neighbors, it tries to activate them in an arbitrary order. Note that a node becomes active once, let us say at step $k$, and try to activate its outgoing neighbors exactly once, at step $k + 1$.

Kemp et al. [7] considered the IM under ICM. They showed that the greedy algorithm works for this model, too. They also demonstrated that it is *NP*-hard to approximate the problem within any factor better than $1 - \frac{1}{e}$.

## 3. Multi-Winner Election Control: Models and Objective Functions

In this section, we consider the Multi-Winner Election Control, where some parties are running for an election so that more than one candidate will be elected as the winner, like a parliament election. We consider $t$ different parties $C_1, \ldots, C_t$, each of them contains $k$ different candidates, i.e., $C_i = \{c_1^i, \ldots, c_k^i\}, 1 \leqslant i \leqslant t$. We use $C$ for the set of all candidates, i.e., $C = \cup_{i=1}^t C_i$. Furthermore, without loss of generality, we assume $C_1$ is our target party. Note that there will be exactly $k$ winners for the election.

### 3.1. Multi-Winner Election Control under LTM

In this model, we investigate the case that the adversary does not know the preferences list of the voters; instead of that, for each voter, the attacker has a probability distribution over all candidates. This model is similar to the model known as probabilistic linear threshold ranking (PLTR) defined in [8]. Since most voters do not reveal their preferences in social media, then it is a realistic assumption.

The adversary tries to maximize/minimize the number of winners in his target party. For each node $v \in V$, we show $\pi_v$ as the probability distribution of the voter/node $v$ over all candidates; we define $\pi_v(c)$ as the probability that the voter $v$ votes for a specific candidate $c \in C$. Then for every node $v \in V$, and candidate $c \in C$ we have $\pi_v(c) \in [0, 1]$, and $\sum_{c \in C} \pi_v(c) = 1$.

In LTM, each node has an incoming influence, which shows the amount of pressure from incoming neighbors to support/oppose a target party. We use this incoming influence of node $v \in V$ to change its probability distribution. Let us define $\tilde{\pi}_v$ as the probability distribution of node $v$ after $S$. Respectively, $\tilde{\pi}_v(c)$ is the probability that node $v$ will vote for candidate $c \in C$ after $S$. We use $A_S$ to show the set of nodes that will become active after $S$.

We consider a single message which spreads among the voters. The message contains some constructive/destructive information targeting all candidates in the target party. When a node $v$ becomes active, its probability distribution will change according to the incoming influence from its activated neighbors. We have to normalize the vector in order to make sure that the sum of the probabilities is equal to one, after $S$. For constructive model the probability distribution of a node $v \in A_S$ changes as follows.

$$\forall c \in C_1 : \tilde{\pi}_v(c) = \frac{\pi_v(c) + \frac{1}{|C_1|} \sum_{u \in A_S \cap N_v^i} b_{uv}}{1 + \sum_{u \in A_S \cap N_v^i} b_{uv}},$$

$$\forall c \in C \setminus C_1 : \tilde{\pi}_v(c) = \frac{\pi_v(c)}{1 + \sum_{u \in A_S \cap N_v^i} b_{uv}}.$$

Recall that $N_v^i$ is the set of incoming neighbors of node $v$. Furthermore, considering the destructive case, the probability distribution of an active node $v \in A_S$ will change as follows.

$$\forall c \in C_1 : \tilde{\pi}_v(c) = \frac{\pi_v(c)}{1 + \sum_{u \in A_S \cap N_v^i} b_{uv}}$$

$$\forall c \in C \setminus C_1 : \tilde{\pi}_v(c) = \frac{\pi_v(c) + \frac{1}{|C \setminus C_1|} \sum_{u \in A_S \cap N_v^i} b_{uv}}{1 + \sum_{u \in A_S \cap N_v^i} b_{uv}}$$

By these changes (and normalization), we guarantee that the sum of the probability for each node is equal to 1. In both constructive and destructive cases, the probability distribution of inactive nodes $v \in V \setminus A_S$ will not change after $S$, i.e., $\tilde{\pi}_v = \pi_v$.

Let us define the expected number of votes for candidate $c \in C$ after $S$, as $\mathcal{F}(c, S) = \mathbb{E}_{A_S}[\sum_{v \in V} \tilde{\pi}_v(c)]$; similarly, $\mathcal{F}(c, \varnothing) = \mathbb{E}[\sum_{v \in V} \pi_v(c)]$ is the expected number of votes for candidate $c \in C$ before any diffusion.

**Example 1.** *Assume there are two parties supporting two candidates each, i.e., $C = C_1 \cup C_2$, $C_1 = \{c_1^1, c_2^1\}, C_2 = \{c_1^2, c_2^2\}$. There are five nodes in the given graph $G = (V, E)$, where their connections form a star and the weight of all edges is one, i.e., $(v_1, v_2), (v_1, v_3), (v_1, v_4), (v_1, v_5) \in E$, $b_{v_1, v_2} = b_{v_1, v_3} = b_{v_1, v_4} = b_{v_1, v_5} = 1$. Let us consider the probability distribution of each node $v \in V$ as $\pi_v = \pi_v(c_1^1), \pi_v(c_2^1), \pi_v(c_1^2), \pi_v(c_2^2)$. We set the probability distribution of all nodes as $\frac{1}{8}, \frac{1}{8}, \frac{3}{8}, \frac{3}{8}$. Then before any diffusion, the candidates' score is*

$$\mathcal{F}(c_1^1, \varnothing) = \mathcal{F}(c_2^1, \varnothing) = \frac{5}{8},$$

$$\mathcal{F}(c_1^2, \varnothing) = \mathcal{F}(c_2^2, \varnothing) = \frac{15}{8},$$

*and none of our target candidates have less score than their opponents. Consider the constructive model in which the adversary's budget is one, i.e., he can select one node to influence the voters and change their opinion. Since the node $v_1 \in V$ is the most influential node in the graph, the adversary selects it as his seed node. It activates all nodes in the graph, and their probability distribution will be updated as follows.*

$$\tilde{\pi}_{v_1} = \frac{1}{8}, \frac{1}{8}, \frac{3}{8}, \frac{3}{8},$$

$$\tilde{\pi}_{v_2} = \tilde{\pi}_{v_3} = \tilde{\pi}_{v_4} = \tilde{\pi}_{v_5} = \frac{5}{16}, \frac{5}{16}, \frac{3}{16}, \frac{3}{16},$$

*and the expected number of votes for the candidates is*

$$\mathcal{F}(c_1^1, S) = \mathcal{F}(c_2^1, S) = \frac{11}{8},$$

$$\mathcal{F}(c_1^2, S) = \mathcal{F}(c_2^2, S) = \frac{9}{8},$$

*and our target candidates' score is more than their opponents' score.*

### 3.2. Multi-Winner Election Control under ICM

Our model is similar to the work presented in [9]. We briefly mention the model bellow. In this model, despite LTM, we assume that the attacker knows the voters' preference list. Each voter $v \in V$ has a preferences list $\pi_v$. Abusing the notations, $1 \leqslant \pi_v(c) \leqslant tk$ is the rank of candidate $c$ in the preference list of the voter $v$. After the diffusion, inactive voters will keep their original opinions, i.e., $\forall v \in V \setminus A_S : \tilde{\pi}_v = \pi_v$; however, the activated voters will change their preferences list as follows. Remind that $A_S$ is the set of activated nodes after $S$.

- Constructive: For each node $v \in A_S$ and for each target candidate $c \in C_1$, the new position of $c$ in $\tilde{\pi}_v$ is

$$\tilde{\pi}_v(c) = \begin{cases} \pi_v(c) - 1 & \text{if } \exists\, c' \in C \setminus C_1 \text{ s.t. } \pi_v(c') < \pi_v(c) \\ \pi_v(c) & \text{otherwise,} \end{cases}$$

also, for other candidates $c \in C \setminus C_1$, if there is a candidate $c' \in C \setminus C_1$ s.t. $\pi_v(c') = \pi_v(c) + 1$, then we set $\tilde{\pi}_v(c) = \pi_v(c)$; otherwise the new rank of the candidate $c$ will be calculated as follows.

$$\tilde{\pi}_v(c) = \pi_v(c) + |\{c'' \in C_1 \mid \pi_v(c'') > \pi_v(c) \wedge (\nexists\, \bar{c} \in C \setminus C_1 : \pi_v(c) < \pi_v(\bar{c}) < \pi_v(c''))\}|\,.$$

- Destructive: For each node $v \in A_S$ and for each target candidate $c \in C_1$, we have

$$\tilde{\pi}_v(c) = \begin{cases} \pi_v(c) + 1 & \text{if } \exists\, c' \in C \setminus C_1 \text{ s.t. } \pi_v(c') > \pi_v(c) \\ \pi_v(c) & \text{otherwise,} \end{cases}$$

while for $c \in C \setminus C_1$, if there exists a candidate $c' \in C \setminus C_1$ s.t. $\pi_v(c') = \pi_v(c) - 1$ we set $\tilde{\pi}_v(c) = \pi_v(c)$, otherwise we have

$$\tilde{\pi}_v(c) = \pi_v(c) - |\{c'' \in C_1 \mid \pi_v(c'') < \pi_v(c) \wedge (\nexists\, \bar{c} \in C \setminus C_1 : \pi_v(c'') < \pi_v(\bar{c}) < \pi_v(c))\}|\,.$$

In this article, we consider the plurality scoring rule for simplicity, where just the most preferred candidate of each voter gets one score. However, the results can be extended for any non-increasing scoring function, e.g., $k$-approval, anti-plurality, and Borda's rule [25]. Let us denote by $\mathcal{F}(c, \varnothing), \mathcal{F}(c, S)$, the expected score of candidate $c$ before and after $S$, respectively; formally, $\forall c \in C : \mathcal{F}(c, \varnothing) = \sum_{v \in V} \mathbb{1}_{\pi_v(c)=1}, \mathcal{F}(c, S) = \mathbb{E}_{A_S}\left[\sum_{v \in V} \mathbb{1}_{\tilde{\pi}_v(c)=1}\right]$. (If we want to generalize the problem and consider any non-increasing scoring function $g(\cdot)$, the functions would be defined as $\mathcal{F}(c, \varnothing) = \sum_{v \in V} g(\pi_v(c)), \mathcal{F}(c, S) = \mathbb{E}_{A_S}\left[\sum_{v \in V} g(\tilde{\pi}_v(c))\right]$).

**Example 2.** *Consider the graph G and candidates C in Example 1. Let set the voters' preference list as follows.*

$$\pi_{v_1} = c_1^1 \succ c_2^1 \succ c_1^2 \succ c_2^2,$$
$$\pi_{v_2} = c_1^2 \succ c_2^1 \succ c_1^1 \succ c_2^2,$$
$$\pi_{v_3} = c_2^2 \succ c_1^2 \succ c_1^1 \succ c_2^1,$$
$$\pi_{v_4} = c_1^2 \succ c_2^1 \succ c_1^1 \succ c_2^2,$$
$$\pi_{v_5} = c_2^2 \succ c_1^1 \succ c_1^2 \succ c_2^1,$$

*where $a \succ b$ means $a$ is preferred to $b$. The candidates' score before any diffusion is*

$$\mathcal{F}(c_1^1, \varnothing) = 1,$$
$$\mathcal{F}(c_2^1, \varnothing) = 0,$$
$$\mathcal{F}(c_1^2, \varnothing) = \mathcal{F}(c_2^2, \varnothing) = 2,$$

*and before any diffusion, both of our target candidates have less score than their opponents. Consider the constructive case where the adversary's budget is one. The same as Example 1, the adversary selects the node $v_1$ as a seed node, and it activates all nodes in the graph. After S, the voters update their preference list as follows.*

$$\pi_{v_1} = c_1^1 \succ c_2^1 \succ c_1^2 \succ c_2^2,$$
$$\pi_{v_2} = c_2^1 \succ c_1^1 \succ c_1^2 \succ c_2^2,$$
$$\pi_{v_3} = c_2^2 \succ c_1^1 \succ c_2^1 \succ c_1^2,$$

$$\pi_{v_4} = c_2^1 \succ c_1^1 \succ c_1^2 \succ c_2^2,$$
$$\pi_{v_5} = c_1^1 \succ c_2^2 \succ c_2^1 \succ c_1^2,$$

*and the candidates' score will be as follows.*

$$\mathcal{F}(c_1^1, S) = \mathcal{F}(c_2^1, S) = 2,$$
$$\mathcal{F}(c_1^2, S) = 0,$$
$$\mathcal{F}(c_2^2, S) = 1,$$

*and both of the target candidates get more vote than their opponents.*

### 3.3. Objective Functions

In this paper, our goal is to maximize/minimize the number of winners from our target party. Then the objective functions are the same as [9]. Considering both IC and LT models, we define $\mathcal{F}(C_1, S)$ as the number of candidates in $C_1$ that are among the winners. Formally, consider a set of given activated nodes $A_S$, which became active after $S$. Let us define $\mathcal{F}_{A_S}(c)$ as the expected number of votes that candidate $c$ will receive while $A_S$ is the set of activated nodes. We set $\mathcal{Y}_{A_S}(c)$ as the number of candidates $c' \in C \setminus \{c\}$ where the expected number of their votes is less than $c$. In order to consider the tie-breaking rule, if $\mathcal{F}_{A_S}(c_i^j) = \mathcal{F}_{A_S}(c_{i'}^{j'})$, then $c_i^j$ has more priority than $c_{i'}^{j'}$ if $j < j'$, or $j = j' \wedge i < i'$. Then $\mathcal{Y}_{A_S}(c)$ is defined as

$$\mathcal{Y}_{A_S}(c_i^j) = \left| \{ c_{i'}^{j'} \in C \mid \mathcal{F}_{A_S}(c_i^j) > \mathcal{F}_{A_S}(c_{i'}^{j'}) \vee (\mathcal{F}_{A_S}(c_i^j) = \mathcal{F}_{A_S}(c_{i'}^{j'}) \wedge (j < j' \vee (j = j' \wedge i < i'))) \} \right|.$$

By this definition, we define $\mathcal{F}(C_1, S)$ as the expected number of winners from party $C_1$, i.e., $\mathcal{F}(C_1, S) = \mathbb{E}_{A_S} \left[ \sum_{c \in C_1} \mathbb{1}_{\mathcal{Y}_{A_S}(c) \geqslant (t-1)k} \right]$.

Now, let us define the first objective function as Difference of Winners (DoW), where is the difference between the number of winners in our target party before and after $S$. Formally, in constructive (resp., destructive) model we define $\text{DoW}_c$ (resp., $\text{DoW}_d$) as

$$\text{DoW}_c(C_1, S) = \mathcal{F}(C_1, S) - \mathcal{F}(C_1, \varnothing),$$
$$\text{DoW}_d(C_1, S) = \mathcal{F}(C_1, \varnothing) - \mathcal{F}(C_1, S).$$

The problem of constructive difference of winners (CDW) asks for finding a set of seed nodes $S$ ($|S| \leqslant B$) to maximize $\text{DoW}_c(C_1, S)$. Similarly, destructive difference of winners (DDW) refers to the problem of finding a set of seed node $S$ ($|S| \leqslant B$) to maximize $\text{DoW}_d(C_1, S)$.

As the second objective function, we define a more compelling one called Margin of Victory (MoV). For constructive case, we define it as DoW plus the difference between the number of winners in the opponent parties with the most winners after and before $S$. Formally, for constructive (resp., destructive) case, we define $\text{MoV}_c$ (resp., $\text{MoV}_d$) as

$$\text{MoV}_c(C_1, S) = \mathcal{F}(C_1, S) - \mathcal{F}(C_A^S, S) - \left( \mathcal{F}(C_1, \varnothing) - \mathcal{F}(C_B, \varnothing) \right),$$
$$\text{MoV}_d(C_1, S) = \mathcal{F}(C_1, \varnothing) - \mathcal{F}(C_B, \varnothing) - \left( \mathcal{F}(C_1, S) - \mathcal{F}(C_A^S, S) \right),$$

where $C_B, C_A^S$, respectively, are the opponent parties with the most winner before and after $S$.

The constructive margin of victory (CMV) problem is looking for a set of seed nodes $S$ ($|S| \leqslant B$) in order to maximize $\text{MoV}_c(C_1, S)$. Similarly, destructive margin of victory (DMV) refers to the problem of finding a set of seed nodes $S$ ($|S| \leqslant B$) to maximize $\text{MoV}_d(C_1, S)$.

## 4. Multi-Winner Election Control on Graph under LTM

It is proven that the problem is *NP*-hard to approximate within any factor of approximation using ICM [9]. In this part, we prove the same statement considering LTM.

**Theorem 1.** *It is NP-hard to approximate* CMV *and* CDW *within any factor on a given graph under LTM.*

**Proof.** Let us reduce the vertex cover (VC) problem to any approximation algorithm for CDW (reps., CMV). In VC, we are given an undirected graph $G = (V, E)$ and an integer $k$; the decision question is: Is there a set of nodes $V' \subseteq V$ ($|V'| \leqslant k$) so that for each edge $(u, v) \in E$, at least one of its vertices are in $V'$? Assume $\mathcal{I}(G, B)$ is a given instance for VC problem, where $G = (V, E)$ is the given graph, and $B$ is an integer value. We create an instance $\mathcal{I}'(G', B)$ for CDW (reps., CMV) so that $G' = (V \cup V' \cup V'', E')$ is the graph build from $G$, and $B$ is also the budget for our problem. Let us consider a case where there are two parties and four candidates, i.e., $t = k = 2, C = C_1 \cup C_2$, $C_1 = \{c_1^1, c_2^1\}, C_2 = \{c_1^2, c_2^2\}$. We fix the order of candidates in the probability distribution of the voter $v$ as $\pi_v = (\pi_v(c_1^1), \pi_v(c_2^1), \pi_v(c_1^2), \pi_v(c_2^2))$, and build $G'$ as follows.

- For each undirected edge $(u, v) \in E$ add two directed edges $(u, v), (v, u)$ to $E'$. Set the weight of each incoming edge to a node $v \in V$ as $\frac{1}{|N_v^i|}$. By this the sum over weight of all incoming edges is equal to one, i.e., $\forall v \in V : \sum_{u \in N_v^i} b_{u,v} = 1$.
- For each node $v \in V$, add two more nodes $v', v''$ to $V', V''$, respectively. Furthermore, add an edge $(v, v')$ to $E'$ with $b_{v,v'} = 1$. Formally, $\forall v \in V : v' \in V', v'' \in V'', (v, v') \in E'$ s.t. $b_{v,v'} = 1$. Note that nodes in $V''$ are isolated.
- Set the preferences list of the nodes as follows.

$$\forall v \in V, \pi_v = (\frac{1}{2}, \frac{1}{2}, 0, 0),$$

$$\forall v' \in V', \pi_{v'} = (\frac{1}{2}, 0, \frac{1}{2}, 0),$$

$$\forall v'' \in V'', \pi_{v''} = (0, 0, \frac{1}{2}, \frac{1}{2}).$$

By this reduction, the score of candidates before any diffusion is $\mathcal{F}(c_1^1, \emptyset) = \mathcal{F}(c_1^2, \emptyset) = |V|$, $\mathcal{F}(c_2^1, \emptyset) = \mathcal{F}(c_2^2, \emptyset) = \frac{1}{2}|V|$. Then $F(C_1, \emptyset) = \mathcal{F}(C_2, \emptyset) = 1$.

Note that in this reduction a node $v$ will become active deterministically, if either it is selected as a seed node, or all of its incoming neighbors are selected as the seed nodes. Then if we can find a set of seed nodes $S \subseteq V$ so that it activates all nodes in $V$ deterministically, the seed set $S$ is also an answer for the corresponding VC problem.

In any approximation algorithm, we know that $S \subseteq V$ after the diffusion; otherwise, if there is a node $v' \in V' \cap S$ we can replace it with its incoming neighbor $v \in V$ such that $(v, v') \in E'$ and we get at least the same value for $\text{MoV}_c, \text{DoW}_c$. Furthermore, if there exists a node $v'' \in V'' \cap S$ one of the following situations holds:

- There exists an inactive node $v \in V \setminus A_S$ after the diffusion $S$. In this case, we can substitute $v$ for $v''$ and then we get at least the same $\text{DoW}_c, \text{MoV}_c$.
- There is no inactive node $v \in V \setminus A_S$. In this case, according to the nodes' probability distribution, when all nodes in $V$ become active, the value of $\text{MoV}_c$ and $\text{DoW}_c$ is maximum. Then even if we remove $v''$ from $S$ it does not change the value of $\text{MoV}_c$ or $\text{DoW}_c$. By the way, in this situation, if there exist any node $v \in V \setminus A_S$ we replace $v''$ with it, otherwise we replace it with a node $v \in V \setminus S$.

Then from now on, we assume $S \subseteq V$.

If all nodes in $V$ become active, since they have an outgoing edge to all nodes $v' \in V'$ with probability one, then all nodes in $V \cup V'$ will become active, and the score of the candidates will be as follows.

$$\mathcal{F}(c_1^1, S) = |V|,$$

$$\mathcal{F}(c_2^1, S) = \mathcal{F}(c_1^2, S) = \frac{3}{4}|V|,$$

$$\mathcal{F}(c_2^2, S) = \frac{1}{2}|V|.$$

Then $F(C_1, S) = 2, \mathcal{F}(C_2, S) = 0, \mathrm{DoW}_c(C_1, S) > 0, \mathrm{MoV}_c(C_1, S) > 0$, and any approximation algorithm will return a positive value, then the answer of $\mathcal{I}$ will be YES.

On the other hand, if there is a node $v \in V$, which is inactive after the diffusion, i.e., $\exists v \in V \setminus A_S$, the score of candidates will be as follows.

$$\mathcal{F}(c_1^1, S) = |V|,$$

$$\mathcal{F}(c_2^1, S) < \frac{3}{4}|V|,$$

$$\mathcal{F}(c_1^2, S) > \frac{3}{4}|V|,$$

$$\mathcal{F}(c_2^2, S) = \frac{1}{2}|V|.$$

Then $F(C_1, S) = \mathcal{F}(C_2, S) = 1, \mathrm{DoW}_c(C_1, S) = \mathrm{MoV}_c(C_1, S) = 0$, and any approximation algorithm will return zero, then the answer of $\mathcal{I}$ will be NO.

For the other direction, note that if we can find a set of nodes $S \subseteq V$, which is an answer for $\mathcal{I}$, using the same set of nodes, we can activate all nodes in $V \cup V'$ and $\mathrm{DoW}_c(C_1, S) > 0, \mathrm{MoV}_c(C_1, S) > 0$.

To extend the proof for any number of parties ($t$) and candidates ($k$), we need to assign the probability distribution as follows, and the same approach concludes the proof for any $t, k > 2$. The same as before, the order of the candidates in probability distribution of a voter $v$ is $\pi_v = (\pi_v(c_1^1), \ldots, \pi_v(c_k^1), \pi_v(c_1^2), \ldots, \pi_v(c_k^2), \ldots, \pi_v(c_1^t), \ldots, \pi_v(c_k^t))$.

$$\forall v \in V, \pi_v = (\overbrace{\frac{1}{k}, \frac{1}{k}, \ldots, \frac{1}{k}}^{k}, \overbrace{0, \ldots, 0}^{k(t-1)}),$$

$$\forall v' \in V', \pi_{v'} = (\overbrace{\frac{1}{k}, \frac{1}{k}, \ldots, \frac{1}{k}}^{k-1}, 0, \frac{1}{k}, \overbrace{0, \ldots, 0}^{k(t-1)-1}),$$

$$\forall v'' \in V'', \pi_{v''} = (\overbrace{0, \ldots, 0}^{k}, \overbrace{\frac{1}{k}, \ldots, \frac{1}{k}}^{k}, \overbrace{0, \ldots, 0}^{k(t-2)}).$$

$\square$

The following theorem proves the same statement for the destructive case of the problem.

**Theorem 2.** *It is NP-hard to approximate* DMV *and* DDW *within any factor on a given graph under LTM.*

**Proof.** The reduction is similar to the constructive case. Consider the case where $t = k = 2$. We should set the voters' probability distributions such that one of our target candidates be among the losers before and after any diffusion. Furthermore, another target candidate is among the winners before any dissemination; however, he will lose the election if and only if all nodes in the connected part of the

graph become active. Please note that, since our target candidates have more priority than the others, we need one more node to be able to do that. □

## 5. Multi-Winner Election Control on Arborescence under ICM

In this section, instead of a general graph, we consider an arborescence structure. We are given a tree $G = (V, E)$ and a budget $B$ where the directed edges are from leaves towards the root under ICM. We are asked to find at most $B$ seed nodes to maximize $\text{MoV}_c$ and $\text{DoW}_c$.

It has been shown that the problem in inapproximable on a general graph, except $P = NP$ [9]. Bharathi et al. conjectured that the IM problem considering ICM on arborescence is *NP*-hard [26]. Lu et al. proved that the conjecture is true [27], while Wang et al. showed that the IM problem accepts a polynomial-time algorithm on arborescence under LTM [28]. In the following, we show that our problem is hard to approximate within any factor of approximation on arborescence under ICM.

**Theorem 3.** *It is NP-hard to find an approximation algorithm for* CMV *and* CDW *on arborescence under ICM.*

**Proof.** We show the hardness by reducing the IM problem to our problem. Given an instance $\mathcal{I}(T, B)$ of IM problem where $T = (V, E)$ is the tree (arborescence), and $B$ is the budget. Let us define the decision version of the problem as follows: is there at most $B$ seed nodes so that it activates all nodes of the tree in expected?

We consider the case where there are two parties and each of them have just two candidates, i.e., $C = C_1 \cup C_2, C_1 = \{c_1^1, c_2^1\}, C_2 = \{c_1^2, c_2^2\}$. Furthermore, for simplicity, we consider the plurality scoring rule. The proof can be extended for any number of parties and candidates using any non-increasing scoring function, akin to [29].

Let us create an instance of our problem $\mathcal{I}'(T', B)$ as follows, where $T' = (V \cup V' \cup V'', E)$ is a tree, and $B$ is the same budget for both problems.

- For each node $v \in V$ we add two more nodes $v', v''$ to $V', V''$, respectively, i.e., $\forall v \in V : v' \in V'$, $v'' \in V''$.
- For each node $v \in V$ we add an edge $(v, v'')$ to $E$ where $b_{v,v''} = 1$.
- Set the preference list of all nodes as follows.

$$\forall v \in V : c_1^2 \succ c_2^2 \succ c_1^1 \succ c_2^1,$$
$$\forall v' \in V' : c_2^2 \succ c_1^2 \succ c_2^1 \succ c_1^1,$$
$$\forall v'' \in V'' : c_1^2 \succ c_1^1 \succ c_2^1 \succ c_2^2$$

Clearly, seed nodes will be selected from $V$, i.e., $S \subseteq V$; otherwise, if there is a node $v' \in S \cap V'$, then the node is useless and does not affect $\text{DoW}_c$ or $\text{MoV}_c$. If there is a node $v'' \in S \cap V''$, we can replace it with its incoming neighbor and get at least the same value for $\text{DoW}_c$ and $\text{MoV}_c$.

Using aforementioned polynomial-time reduction, if there exists a set of nodes $S \subseteq V$ ($|S| \leqslant B$) so that $\text{MoV}_c > 0$ (resp. $\text{DoV}_c > 0$), then the node will activate all nodes in $V \cup V''$. Hence, we can select the same set and they will activate all nodes in $T$; then the answer of $\mathcal{I}$ will be YES. On the other hand, if $\text{MoV}_c = 0$ (resp. $\text{DoW}_c = 0$), it means there is no seed set can activate all nodes in $V \cup V''$; then the answer of $\mathcal{I}$ is NO. More formally, before any diffusion the score of candidates is

$$\mathcal{F}(c_1^1, \varnothing) = \mathcal{F}(c_2^1, \varnothing) = 0,$$
$$\mathcal{F}(c_1^2, \varnothing) = 2|V|,$$
$$\mathcal{F}(c_2^2, \varnothing) = |V|.$$

Then, none of the candidates in our target party will be elected as winner. After $S$, if there exists an inactive node in $V \cup V''$, then the the score of candidates will be as follows:

$$\mathcal{F}(c_1^1, S) < |V|,$$
$$\mathcal{F}(c_2^1, S) = 0,$$
$$\mathcal{F}(c_1^2, S) > |V|,$$
$$\mathcal{F}(c_2^2, S) = |V|.$$

In this case also, none of our target candidates will be among the winners, and $\mathrm{MoV}_c = \mathrm{DoW}_c = 0$. However, if all nodes in $V \cup V''$ become active after $S$, the score of the candidates will be as follows and one of our target candidates ($c_1^1$) will be elected as winner and any approximation algorithm will return $\mathrm{MoV}_c > 0$ (resp. $\mathrm{DoW}_c > 0$). It concludes the prove.

$$\mathcal{F}(c_1^1, S) = |V|,$$
$$\mathcal{F}(c_2^1, S) = 0,$$
$$\mathcal{F}(c_1^2, S) = |V|,$$
$$\mathcal{F}(c_2^2, S) = |V|.$$

□

The following theorem demonstrates the same hardness of approximation for the destructive case of our problem.

**Theorem 4.** *It is NP-hard to find an approximation algorithm for* DMV *and* DDW *on arborescence under ICM.*

**Proof.** The prove for the destructive case is similar to the constructive one. Consider $\mathcal{I}'$ in Theorem 3, we need to set the preferences list of the nodes so that all of our target candidates win the election before any diffusion; however, after the diffusion, one of them (let us say $c \in C_1$) will lose if and only if all nodes in $V \cup V''$ become active. Note that since our target candidates have more priority than the others, we need one more isolated node to ensure that $c$ will lose the election after the diffusion. Following the same approach concludes the statement. □

## 6. Multi-Winner Election Control on Tree Using Straight-Party Voting

In this part, we consider the problem on a variation of the straight-party voting system (also called straight-ticket voting) in which the voters can vote for a party instead of candidates [30,31]. This model is used in many real elections [32,33]. The multi-winner election control problem via social influence under ICM and a general graph is considered in [9]. They showed that the problem is hard, and presented some constant factor approximation using straight-party voting system. In this section, we consider the problem on a tree where the edges are directed from root to the leaves.

In the rest of this section, we assume the given tree is a binary tree as we can convert any tree $T$ to a binary tree $T'$ by adding $O(n)$ fake nodes. However, our algorithm can use the fake nodes to navigate the tree, but they neither have a probability distribution (preference list) nor can be selected as a seed node. To ensure that the fake nodes will not change the diffusion process on the tree, the weight of each incoming edge to each fake node should be equal to one. Moreover, the weight of an edge from a fake node to an original node is equal to the weight of the original node's incoming edge in $T$.

In the following, we present some dynamic programming (DP) algorithm to maximize $\mathrm{DoV}_c^{spv}$ (and $\mathrm{DoV}_d^{spv}$). Given a tree $T = (V, E)$, and budge $B$, the idea is that for a fixed node $v \in V$ and budget $k$ ($0 \leqslant k \leqslant B$), we calculate the maximum outcome from the sub-tree rooted at $v$, among the following cases: First, select the node $v$ and try to find the other $k - 1$ seed nodes in its children. Second, do not select $v$ and look for $k$ seed nodes in its children.

We define $r(v), l(v), f(v)$, respectively, as the right child, left child, and the parent (father) of the node $v$. In Section 6.1 we consider the problem under LTM, and in Section 6.2 the problem is investigated under ICM.

*6.1. Multi-Winner Election Control Using Straight-Party Voting under LTM*

In this section, the voters have preferences list over the candidates. However, they vote for a party proportional to the probability of voting for all candidates in each party. Let us define $\mathcal{F}_{spv}(C_1, \varnothing), \mathcal{F}_{spv}(C_1, S)$, as the sum of the scores for our target party $C_1$ before and after $S$, respectively. Formally they are defined as follows.

$$\mathcal{F}_{spv}(C_1, \varnothing) = \mathbb{E}\Big[\sum_{v \in V} \sum_{c \in C_1} \pi_v(c)\Big],$$

$$\mathcal{F}_{spv}(C_1, S) = \mathbb{E}_{A_S}\Big[\sum_{v \in V} \sum_{c \in C_1} \tilde{\pi}_v(c)\Big].$$

The same as before we define the objective function MoV and difference of votes (DoV), for constructive case, as follows.

$$\mathrm{DoV}_c^{spv}(C_1, S) = \mathcal{F}_{spv}(C_1, S) - \mathcal{F}_{spv}(C_1, \varnothing),$$
$$\mathrm{MoV}_c^{spv}(C_1, S) = \mathcal{F}_{spv}(C_1, S) - \mathcal{F}_{spv}(C_A^S, S) - \big(\mathcal{F}_{spv}(C_1, \varnothing) - \mathcal{F}_{spv}(C_B, \varnothing)\big), \tag{1}$$

while $C_B$ and $C_A^S$ are the most voted opponent party before and after $S$, respectively. For destructive model the objective functions are defined as

$$\mathrm{DoV}_d^{spv}(C_1, S) = \mathcal{F}_{spv}(C_1, \varnothing) - \mathcal{F}_{spv}(C_1, S),$$
$$\mathrm{MoV}_d^{spv}(C_1, S) = \mathcal{F}_{spv}(C_1, \varnothing) - \mathcal{F}_{spv}(C_B, \varnothing) - \big(\mathcal{F}_{spv}(C_1, S) - \mathcal{F}_{spv}(C_A^S, S)\big). \tag{2}$$

6.1.1. Maximizing DoV in Straight-Party Voting under LTM

We define $F_v$ as the set of possible probabilities that the node $f(v)$ may become active. More precisely, consider all nodes in the path from root to the $v$ as $F_v' = \{v_0, v_1, \ldots, v_t = f(v)\}$ (recall that $f(v)$ is the parent of $v$). If none of the nodes in $F_v'$ are selected as a seed node, then the probability that $f(v)$ becomes active by his incoming influence is zero. If just the root ($v_0$) is selected as the seed node, then the probability that $f(v)$ becomes active is $\prod_{i=0}^{i<t} b_{v_i, v_{i+1}}$; also, if $v_1$ is selected as a seed node but none of the nodes $v_i, 2 \leqslant i \leqslant t$, are selected as a seed node, the probability that $f(v)$ becomes active by its parent is $\prod_{i=1}^{i<t} b_{v_i, v_{i+1}}$, and so on; all these probabilities belong to $F_v$.

Let us define $\mathrm{DoV}_c(v, k, S, p)$ as the maximum value of the sum over the difference of probability to vote for our target party after and before $S$ in the sub-tree rooted at $v$ while $p \in F_v$ is the probability that its parent is active, and the budget is $k$. Furthermore, all selected seed nodes will be in $S$. In other words, $\mathrm{DoV}_c(v, k, S, p) = max\{\mathrm{DoV}_c^{spv}(C_1, S)\}$ in the sub-tree rooted at $v$ while it will become active with probability $p \cdot b_{f(v),v}$ and $|S| \leqslant k$. The formal definition of $\mathrm{DoV}_c(v, k, S, p)$ is as follows:

$$\mathrm{DoV}_c(v, k, S, p) = max\Bigg\{$$

$$max_{k'=0}^{k}\Big\{\mathrm{DoV}_c\Big(r(v), k', S, p \cdot b_{f(v),v}\Big) + \mathrm{DoV}_c\Big(l(v), k - k', S, p \cdot b_{f(v),v}\Big)\Big\} + p \cdot b_{f(v),v} \cdot \mathcal{D}_v,$$

$$max_{k'=0}^{k-1}\Big\{\mathrm{DoV}_c\big(r(v), k', S \cup \{v\}, 1\big) + \mathrm{DoV}_c\big(l(v), k - k' - 1, S \cup \{v\}, 1\big)\Big\} + \mathcal{D}_v\Bigg\}, \tag{3}$$

where $\mathcal{D}_v$ is the increased score of our target party made by the node $v$ if it becomes active, which is

$$\mathcal{D}_v = \sum_{c \in C_1}\left(\frac{\pi_v(c) + \frac{1}{|C_1|} \cdot p \cdot b_{f(v),v}}{1 + p \cdot b_{f(v),v}} - \pi_v(c)\right). \tag{4}$$

We can calculate and store the values in a two-dimensional array $A[B+1,|V|]$ where the rows are the budgets (starting from zero to $B$), and the columns are the nodes of the tree presented as the BFS reverse order, and each cell $(i,j)$ ($0 \leqslant i \leqslant B, 0 \leqslant j < |V|$) of the array refers to another array $A'[|F_{v_j}|]$. Then in the worst case, since the budget $B$, and $|F_{v_j}|$ (for any $v_j \in V$) are at most equal to $|V|$, then we can solve the problem in polynomial time using $O(|V|^3)$ memory. Note that we have to fill the matrix $A$ left-to-right and top-down, while for each cell of it we can fill the corresponding array $A'$ in any order.

As the base cases, for each leaf $v \in V$, and $p \in F_v$, if $k > 0$ we set $\mathrm{DoV}_c(v,k,S,p) = \mathcal{D}_v$, otherwise, if $k = 0$ we have $\mathrm{DoV}_c(v,k,S,p) = p \cdot b_{f(v),v} \cdot \mathcal{D}_v$ which is the difference of the probability to vote for our party after and before diffusion $S$, made by the node $v$. In fact, if the budget is greater than zero, the node will become active for sure, and we need to consider the difference of scores, but if the budget is zero we cannot select it as a seed node and the value should be multiplied by the probability that the node will become active, i.e., $p \cdot b_{f(v),v}$. We also define $\mathrm{DoV}_c(null, k, S, p) = 0$, that is, the value of $\mathrm{DoV}_c$ for a null reference is zero. It is useful when a node has just left (resp. right) child, then the value of the function for its right (resp. left) child, regardless of the other parameters, is zero. The pseudo-code of the DP is presented in Algorithm 1, which calculates the maximum $\mathrm{DoV}_c^{spv}$; by small changes, it can find the seed nodes too. Note that the final answer will be calculated by $\mathrm{DoV}_c(v_{root}, B, \varnothing, 0)$ where $v_{root}$ is the root node of the tree, $B$ is the budget, $\varnothing$ represents that we have no seed node so far, and $0$ means the parent of the root node will be activated with zero probability. The following theorem shows that the DP works well.

---

**Algorithm 1**: Calculating maximum $\mathrm{DoV}_c$ for e given tree $T$ and budget $B$ when the diffusion model is LTM and voting system is straight-party voting.

---

> **Procedure** *DoV(Tree $T = (V, E)$, Budget $B$)*
> > $A \leftarrow [B+1, |V|]$        ▷ `It is a two-dimensional array` $A[0..B, 0..|V|-1]$
> > Name all nodes in $V$ from $0$ to $|V|-1$ in BFS reverse order
> > **for** *($j \leftarrow 0; j < |V|; j \leftarrow j+1$)* **do**
> > > $F_{v_j} \leftarrow$ Set of all possible probabilities that $f(v_j)$ may become active
> > > **for** *($i \leftarrow 0; i <= B; i \leftarrow i+1$)* **do**
> > > >      ▷ `the variables` $i, j$ `are a counter for rows and columns, respectively.`
> > > > $A[i,j] \leftarrow \mathrm{Array}[|F_{v_j}|]$       ▷ `Each cell` $(i,j)$ `is an array`
> > > > **if** *($v_j$ is a leaf)* **then**
> > > > > **for** *($p \in F_{v_j}$)* **do**
> > > > > > $A[i,j;p] \leftarrow \sum_{c \in C_1} \left( \frac{\pi_{v_j}(c) + \frac{1}{|C_1|} \cdot p \cdot b_{f(v_j),v_j}}{1 + p \cdot b_{f(v_j),v_j}} - \pi_{v_j}(c) \right)$
> > > > > > **if** *($i = 0$)* **then**
> > > > > > > $A[i,j;p] \leftarrow p \cdot b_{f(v_j),v_j} \cdot A[i,j;p]$
> > > > > > **end**
> > > > > **end**
> > > > **end**
> > > > **continue**
> > > > **end**
> > > > **for** *($p \in F_{v_j}$)* **do**
> > > > >      ▷ `If` $r(v_j)$ `or` $l(v_j)$ `does not exist,` $A[\ldots, r(v_j)$ `or` $l(v_j); \ldots]$ `is zero.`
> > > > > $\mathcal{D}_v \leftarrow \sum_{c \in C_1} \left( \frac{\pi_{v_j}(c) + \frac{1}{|C_1|} \cdot p \cdot b_{f(v_j),v_j}}{1 + p \cdot b_{f(v_j),v_j}} - \pi_{v_j}(c) \right)$
> > > > > $max_j \leftarrow \max_{k=0}^{i}(A[k, r(v_j); p \cdot b_{f(v_j),v_j}] + A[i-k, l(v_j); p \cdot b_{f(v_j),v_j}])$
> > > > > $max'_j \leftarrow \max_{k=0}^{i-1}(A[k, r(v_j); 1] + A[i-k-1, l(v_j); 1])$
> > > > > $A[i,j;p] \leftarrow \max(max_j + p \cdot b_{f(v_j),v_j} \cdot \mathcal{D}_v, max'_j + \mathcal{D}_v)$
> > > > **end**
> > > **end**
> > **end**
> > **return** $A[B, |V|-1; 0]$       ▷ `The final result for the root node using all budget`
> **end**

---

**Theorem 5.** *Given a tree $T = (V, E)$ and budget $B$, the DP Equation (3) finds a set of seed nodes $S$ ($|S| \leqslant B$) to maximize $\text{DoV}_c^{spv}$.*

**Proof.** Consider the matrix $A[B + 1, |V|]$ where each cell $A[k, v]$ point to another array $A'$ where the columns are all possible probabilities that $f(v)$ will become active. Calculating all possible probabilities for the array $A'$, we have at most $|F_v|$ columns for each node $v \in V$ and budget $0 \leqslant k \leqslant B$, and for each of them, we need to calculate and store the maximum $\text{DoV}_c$.

Please note that if $f(v)$ becomes active, it can activate $v$ with a probability equal to the weight of the edge between them ($b_{f(v),v}$). It holds because each node has just one incoming edge (its parent), and the threshold of the node will be generated uniformly at random. Then the probability that the threshold of the node $v$ be less than (or equal) to the weight of the incoming edge is $b_{f(v),v}$.

Let us show that all values in the arrays will be calculated correctly, by induction. To see that, consider the base cases. For each leaf $v \in V$, the node cannot activate any other node as it has no outgoing edge. Then, these nodes cannot change the probability distribution of other nodes. In other words, each leaf will change just its own probability distribution. If $k = 0$, it means that we cannot select the node as a seed node, and we need to consider the probability of activating the node, because just activated nodes can update their probability distribution after the diffusion. Then if $k = 0$, we have $\text{DoV}_c(v, k, S, p) = p \cdot b_{f(v),v} \cdot \mathcal{D}_v$, where $\mathcal{D}_v$ is the difference of the party's score if the node $v$ becomes active (defined in Equation (4)), and $p \cdot b_{f(v),v}$ is the probability that the node will be activated by its parent. On the other hand, if $k > 0$, we can select $v$ as a seed node, and it will be activated with the probability of one, then we have $\text{DoV}_c(v, k, S, p) = \mathcal{D}_v$. Using the updating rule (defined in Section 3.1), and the definition of $\text{DoV}_c^{spv}$ (defined in Equation (1)), the base cases are true.

Let us define $(i', j') < (i, j)$ if $j' < j$, or $j' = j \wedge i' < i$. We have shown that all arrays $A'$ related to the base cases filled out correctly. Now by induction step, assume all related arrays related to pair $(i', j')$ smaller than $(i, j)$ are correctly calculated. In order to calculate the $A'$ related to $A[i, j]$, for each column $p \in F_{v_j}$ we use following formula

$$
\text{DoV}_c(v_j, i, S, p) = max \Bigg\{
$$
$$
max_{k=0}^{i} \Big\{ \text{DoV}_c \left( r(v_j), k, S, p \cdot b_{f(v_j),v_j} \right) + \text{DoV}_c \left( l(v_j), i - k, S, p \cdot b_{f(v_j),v_j} \right) \Big\} + p \cdot b_{f(v_j),v_j} \cdot \mathcal{D}_{v_j},
$$
$$
max_{k=0}^{i-1} \Big\{ \text{DoV}_c \left( r(v_j), k, S \cup \{v_j\}, 1 \right) + \text{DoV}_c \left( l(v_j), i - k - 1, S \cup \{v_j\}, 1 \right) \Big\} + \mathcal{D}_{v_j} \Bigg\},
$$

in which the first maximization considers the maximum value among all possible cases that we do not select the node $v_j$ as a seed node, and the second one considers the maximum value among all possible cases that we choose $v_j$ as a seed node. The last term in each maximization is the increased amount of $\text{DoV}_c$ in the node $v_j$, which is according to the probability that $v_j$ will become active. Note that in the above formula, we are using the value of $\text{DoV}_c$ for the children of $v_j$, and the nodes are sorted as the BFS reverse order, then all required values are correctly calculated before, and we are selecting the maximum value among all possible cases. Then $\text{DoV}_c(v_j, i, S, p)$ will find the maximum possible value of $\text{DoV}_c^{spv}$ correctly and concludes the proof. $\square$

For the destructive model, we define $\text{DoV}_d(v, k, S, p)$ as the maximum difference of probability to vote for our target party before and after $S$ in the sub-tree rooted at $v$, while the budget is $k$ and $p \in F_v$ is the probability that $f(v)$ will become active. Formally, we define $\text{DoV}_d(v, k, S, p)$ as follows.

$$\text{DoV}_d(v,k,S,p) = max \Bigg\{$$

$$max_{k'=0}^{k}\Big\{\text{DoV}_d\left(r(v),k',S,p\cdot b_{f(v),v}\right) + \text{DoV}_d\left(l(v),k-k',S,p\cdot b_{f(v),v}\right)\Big\} + p\cdot b_{f(v),v}\cdot \mathcal{D}'_v,$$

$$max_{k'=0}^{k-1}\Big\{\text{DoV}_d\left(r(v),k',S\cup\{v\},1\right) + \text{DoV}_d\left(l(v),k-k'-1,S\cup\{v\},1\right)\Big\} + \mathcal{D}'_v\Bigg\}, \quad (5)$$

where $\mathcal{D}'_v = \sum_{c\in C_1}\left(\pi_v(c) - \frac{\pi_v(c)}{1+p\cdot b_{f(v),v}}\right)$ is the difference that the node $v$ can apply. Moreover, for the base cases of the problem, for each leaf $v\in V$, and each probability $p\in F_v$, if $k=0$ we need to consider the probability that the node will become active, then $\text{DoV}_d(v,k,S,p) = p\cdot b_{f(v),v}\cdot \mathcal{D}'_v$; otherwise, if $k>0$, we have $\text{DoV}_d(v,k,S,p) = \mathcal{D}'_v$. Furthermore, we set $\text{DoV}_c(null,k,S,p) = 0$. The same as constructive case, for implementation we need a tow-dimensional array $A[B+1,|V|]$. Moreover, for each cell $(i,j), 0\leqslant i\leqslant B, 0\leqslant j<|V|$, we keep another array $A'[|F_{v_j}|]$, where $F_{v_j}$ is the set of possible probabilities that the node $f(v_j)$ can become active. The following theorem shows that by filling the matrix $A$ left-to-right and up-down direction, we can find the optimal answer for $\text{DoV}_d^{spv}$.

**Theorem 6.** *Given a tree $T = (V,E)$ and a budget B, using the DP Equation (5), we can find a set of seed nodes S ($|S|\leqslant B$) to maximize $\text{DoV}_d^{spv}$.*

**Proof.** The proof is similar to Theorem 5, except for the base cases and the way of updating each activated node's probability distribution after the diffusion. Since a leaf cannot activate any other node, the only change that it can make is updating its own probability distribution. According to the updating rule (in Section 3.1), and the definition of $\text{DoV}_d^{spv}$ (defined in Equation (2)), the base cases hold. Furthermore, by induction, we can see that the DP Equation (5) will find the maximum value of $\text{DoV}_d^{spv}$ correctly. □

### 6.1.2. Maximizing MoV in Straight-Party Voting under LTM

In order to maximize $\text{MoV}_c^{spv}$ we have to know $C_A^S$, i.e., the most voted opponent party after $S$. We have no problem to find the most voted opponent party before any diffusion ($C_B$); however, to find the most voted opponent party after $S$ we need to have the optimal set of seed nodes that maximizes $\text{MoV}_c^{spv}$, and to find the optimal set of seed nodes we need the most voted opponent party (parties), which is a defective cycle.

To deal with this problem, someone may say that we consider $C_i, 2\leqslant i\leqslant t$ as the most voted opponent party after $S$, and solve the related DP; after finding the outcome for all $t-1$ parties, we select the maximum result as the output. Nevertheless, this is not true in all cases. Consider a case that there are two opponent parties, and each of them has half of the votes before any diffusion. If we consider each of them as the most voted opponent after the diffusion, we will get a wrong outcome as they both can be the most voted opponent after different diffusion processes. In fact, we need to consider multiple parties as the most voted opponent party.

By the way, it has been shown that by maximizing $\text{DoV}_c^{spv}$ we get a $\frac{1}{3}$-approximation factor for maximizing $\text{MoV}_c^{spv}$. Moreover, by maximizing $\text{DoV}_d^{spv}$ we get a $\frac{1}{2}$-approximation answer for maximizing $\text{MoV}_d^{spv}$ [8].

### 6.2. Multi-Winner Election Control Using Straight-Party Voting under ICM

As we saw in previous section (in LTM), each node $v$ becomes active either by being among the seed nodes or by the incoming influence from its parent $f(v)$. Since there is just one incoming edge for each node $v\in V$, and the threshold of the nodes $t_v$ is generated uniformly at random, then the

probability that its threshold be less than or equal to the incoming weight ($b_{f(v),v}$) is equal to $b_{f(v),v}$. In other words, the node will become active from its parent with the probability that its parent $f(v)$ is active, times the weight of the edge between them. On the other side, in ICM, a node $v$ becomes active if it is either selected as a seed node or its parent $f(v)$ is activated and tries to influence $v$ with the probability $b_{f(v),v}$. Then in a tree, the activation processes in both LTM and ICM are the same.

However, the updating rule is entirely different in them. In other words, in LTM, voters have a probability distribution over the candidates, and the activated nodes will update the probability of voting for candidates regarding the influence from activated incoming neighbors, while in ICM, voters have an exact preferences list over candidates, and the activated nodes promote/demote the position of some candidates in their preference list, regardless of neighbors (see Section 2 for a formal definition).

Since the diffusion process in ICM is the same as LTM, we focus more on updating part of the problem to maximize $\text{DoV}_c^{spv}$. Recall that we consider the plurality scoring rule for simplicity; however, it is possible to extend the results to any non-increasing scoring function. Then the scoring function $\mathcal{F}_{spv}$ for our target party is defined as follows. (To extend the result using any non-increasing scoring function $g(\cdot)$, we should define the functions as $\mathcal{F}_{spv}(C_1, \emptyset) = \sum_{v \in V} \sum_{c \in C_1} g(\pi_v(c))$, $\mathcal{F}_{spv}(C_1, S) = \mathbb{E}_{A_S}\left[\sum_{v \in V} \sum_{c \in C_1} g(\tilde{\pi}_v(c))\right]$.)

$$\mathcal{F}_{spv}(C_1, \emptyset) = \sum_{v \in V} \sum_{c \in C_1} \mathbb{1}_{\pi_v(c)=1},$$

$$\mathcal{F}_{spv}(C_1, S) = \mathbb{E}_{A_S}\left[\sum_{v \in V} \sum_{c \in C_1} \mathbb{1}_{\tilde{\pi}_v(c)=1}\right],$$

and the objective functions for the constructive and destructive cases of our problem are the same as Equations (1) and (2), respectively.

### 6.2.1. Maximizing DoV in Straight-Party Voting under ICM

In this case, node $v$ can increase our target party's score by one, if none of our target candidates are in the first position before any diffusion, and one of them is in the second position of the voter's preference list. In other words, the voter $v$ may increase the score of our target party if $\exists c \in C_1$, $\exists c' \in C \setminus C_1 : \pi_v(c') = 1 \wedge \pi_v(c) = 2$; otherwise, the node $v$ can influence its children and change their opinion, but it cannot affect the target party's score. We call this condition as pre-condition and show it by $\P_v$. We define $F_v$ as the set of all possible probabilities that the node $v$ may become active (Please note that the definition of $F_v$ in ICM is different from LTM). Consider a sub-tree rooted at $v \in V$, budget $k$, seed set $S$, and $p \in F_v$, we define $\text{DoV}'_c(v, k, S, p)$ as follows.

$$\text{DoV}'_c(v, k, S, p) = max\Big\{$$
$$max_{k'=0}^{k}\{\text{DoV}'_c(r(v), k', S, p \cdot b_{v,r(v)}) + \text{DoV}'_c(l(v), k - k', S, p \cdot b_{v,l(v)})\} + p \cdot \mathbb{1}_{\P_v},$$
$$max_{k'=0}^{k-1}\{\text{DoV}'_c(r(v), k', S \cup \{v\}, b_{v,r(v)}) + \text{DoV}'_c(l(v), k - k' - 1, S \cup \{v\}, b_{v,l(v)})\} + \mathbb{1}_{\P_v}\Big\}. \quad (6)$$

As the base cases of the problem, for each leaf $v \in V$, budget zero, and $p \in F_v$ as the probability that $v$ will become active, we set $\text{DoV}'_c(v, k, S, p) = p \cdot \mathbb{1}_{\P_v}$, and for the same parameters except a budget $k > 0$ we set $\text{DoV}'_c(v, k, S, p) = \mathbb{1}_{\P_v}$. (To extend the algorithm for any non-increasing scoring function $g(\cdot)$, we need to define the base cases, respectively, as $\text{DoV}'_c(v, k, S, p) = p \cdot \left(\sum_{c \in C_1, \exists c' \in C \setminus C_1 : \pi_v(c') < \pi_v(c)} g(\pi_v(c) - 1) - g(\pi_v(c))\right)$ and $\text{DoV}'_c(v, k, S, p) = \sum_{c \in C_1, \exists c' \in C \setminus C_1 : \pi_v(c') < \pi_v(c)} g(\pi_v(c) - 1) - g(\pi_v(c))$.) The same as before, for each reference to a node which does not exists (*null*), we define $\text{DoV}'_c(null, k, S, p) = 0$. In order to implement the DP Equation (6), the idea is the same as Algorithm 1. The following theorem shows that it calculates the maximum $\text{DoV}_c^{spv}$ in polynomial-time.

**Theorem 7.** *Given a tree $T = (V, E)$, and budget B, the DP Equation (6) gives a set of seed nodes S ($|S| \leqslant B$) which maximizes $DoV_c^{spv}$.*

**Proof.** In DP Equation (6), there is a maximization over two other maximization formulae. The first one considers the case that we do not select $v$ as a seed node; in this case, we consider the probability that node $v$ will become active, i.e., $p \in F_v$. The second maximization considers selecting $v$ as a seed node; in this state, $v$ will be activated with probability equal to one. In both cases, the node may increase the function's value if the pre-condition holds; otherwise, it can influence its children. The same as previous proves, we show that it works by induction.

Consider a two-dimensional array $A[B + 1, |V|]$ where rows are the budgets from zero to $B$, and columns are the nodes in BFS reveres order. Each cell $A[i, j]$ ($0 \leqslant i \leqslant B, 0 \leqslant j < |V|$) refers to another array $A'$ with the size of $|F_{v_j}|$. We calculate each array related to each cell $(i, j)$ left-to-right and up-down direction.

To show that the base cases are correct, note that the leaves cannot activate any other node. Their only effect is by becoming active and changing their own opinion. Then there are two cases if the pre-condition holds for a leaf $v$: First, the budget is more than zero, then $v$ can be a seed node and increase the amount of $DoV_c'$ by one. Second, if the budget is zero, $v$ can increment $DoV_c'$ with the probability of becoming active through its parent, i.e., in expected, it will be $p \cdot \mathbb{1}_{\P_v}$ where $p \in F_v$ is the probability that $v$ will be activated through its parent. Note that if the pre-condition does not hold, the leaf cannot make any effect, and in both cases, its effect is equal to zero.

Let us say $(i', j') < (i, j)$ if $j' < j$, or $j' = j \wedge i' < i$. As the step of induction, assume that all cells $(i', j')$ smaller that $(i, j)$ are filled correctly for $0 \leqslant i \leqslant B, 0 \leqslant j < |V|$. In order to calculate the array $A'$ related to the cell $(i, j)$, for each $p \in F_{v_j}$ we have to calculate the result of the following function.

$$
\begin{aligned}
\mathrm{DoV}_c'(v_j, i, S, p) = max\Big\{ \\
max_{k=0}^{i}\{\mathrm{DoV}_c'(r(v_j), k, S, p \cdot b_{v_j, r(v_j)}) + \mathrm{DoV}_c'(l(v_j), i - k, S, p \cdot b_{v_j, l(v_j)})\} + p \cdot \mathbb{1}_{\P_v}, \\
max_{k=0}^{i-1}\{\mathrm{DoV}_c'(r(v_j), k, S \cup \{v_j\}, b_{v_j, r(v_j)}) + \mathrm{DoV}_c'(l(v_j), i - k - 1, S \cup \{v_j\}, b_{v_j, l(v_j)})\} + \mathbb{1}_{\P_v} \Big\}.
\end{aligned}
$$

There is a maximization over two cases. Let us check each case separately. The first case considers all possible cases to split the budget into two parts for its children $r(v_j)$ and $l(v_j)$ (the first and second terms) when $v_j$ is not selected as a seed node. It finds the split with the maximum outcome using the $DoV_c'$ of its children, which are calculated correctly. In this case, since the node $v_j$ is not a seed node, then the probability that its right (resp. left) child will become active is $p \cdot b_{v_j, r(v_j)}$ (resp. $p \cdot b_{v_j, l(v_j)}$). The fixed-term is the amount of change that the node $v_j$ can afford to maximize our target party's score. If the pre-condition holds, then with the probability of $p$ it will increase the score by one, that is $p \cdot \mathbb{1}_{\P_v}$.

The second maximization investigates the same situation except that it selects $v_j$ as a seed node (if $i > 0$) and uses the value $DoV_c'$ of its children to find the best split for the $i - 1$ remaining budgets. In this case, the node $v_j$ can increase our party's score by one (if the pre-condition holds) as it is selected as a seed node and will be activated for sure. (To generalize the proof using any non-increasing scoring function $g(\cdot)$, we should change the updating part of each maximization (the fixed part) as $p \cdot (\sum_{c \in C_1, \exists c' \in C \setminus C_1 : \pi_v(c') < \pi_v(c)} g(\pi_v(c) - 1) - g(\pi_v(c)))$ and $\sum_{c \in C_1, \exists c' \in C \setminus C_1 : \pi_v(c') < \pi_v(c)} g(\pi_v(c) - 1) - g(\pi_v(c))$, respectively.) Note that all corresponding values for the children of $v_j$ are correctly calculated before because the nodes are sorted as BFS reverse order. Finally, it finds the maximum value among the two cases. $\square$

For the destructive case of the problem, we define pre-condition $\P_v'$ as $\exists c \in C_1 : \pi_v(c) = 1$. Then for a node $v$, if it becomes active and $\P_v'$ holds, the node will decrease the party's score by one; otherwise, $v$ cannot change it. For each sub-tree rooted at $v$, budget $k$, and $p \in F_v$, let us define $DoV_d'(v, k, S, p)$ as follows.

$$\mathrm{DoV}'_d(v,k,S,p) = max\Big\{$$
$$max^k_{k'=0}\{\mathrm{DoV}'_d(r(v),k',S,p \cdot b_{v,r(v)}) + \mathrm{DoV}'_d(l(v),k-k',S,p \cdot b_{v,l(v)})\} + p \cdot \mathbb{1}_{\mathbb{q}'_v},$$
$$max^{k-1}_{k'=0}\{\mathrm{DoV}'_d(r(v),k',S \cup \{v\},b_{v,r(v)}) + \mathrm{DoV}'_d(l(v),k-k'-1,S \cup \{v\},b_{v,l(v)})\} + \mathbb{1}_{\mathbb{q}'_v}\Big\}. \quad (7)$$

Note that the definition is exactly the same as constructive case except for the pre-condition. Furthermore the base cases are the same as before if we substitute $\mathbb{q}'_v$ for $\mathbb{q}_v$. The prove of the following theorem is similar to the Theorem 7; then we omit it to avoid repetition.

**Theorem 8.** *Given a tree $T = (V,E)$, and budget B, the DP Equation (7) gives a set of seed nodes S ($|S| \leqslant B$) which maximizes $\mathrm{DoV}^{spv}_d$.*

### 6.2.2. Maximizing MoV in Straight-Party Voting under ICM

Similar to Section 6.1.2, we do not know the most scored parties after the diffusion started from a set of optimal seed nodes. However, it has been shown that by maximizing $\mathrm{DoV}^{spv}_c$ (resp. $\mathrm{DoV}^{spv}_d$) we get a $\frac{1}{3}$ (resp. $\frac{1}{2}$) approximation algorithm for maximizing $\mathrm{MoV}^{spv}_c$ (resp. $\mathrm{MoV}^{spv}_d$) [9].

## 7. Discussion

Controlling election via social influence is one of the most crucial parts of each democratic election. It has been shown that many campaigns are using this powerful tool to influence the voters and change their opinion during elections. In this work, we considered the multi-winner election control utilizing social influence so that the attacker tries to maximize/minimize the number of winners from his target party, concerning the party with the most winners.

We exhibited different results, including hardness of approximation, approximation guarantee, and optimal solutions for our problem considering different structures, diffusion models, and voting systems. In ICM, each voter has a preference list over the candidates and will vote for one or more candidate according to the voting rule, e.g., plurality, Borda's rule, *k*-approval, and anti-plurality. In this case, the influenced voters change their opinion by promoting/demoting the candidates' position in their preference list. On the other hand, in LTM, we consider that the voters have a probability distribution over all candidates. Each voter votes for one or more candidates proportional to the probability of voting for them. In this model, the activated voters change their opinion based on the incoming activated neighbors' influence.

We proved the problem is hard to approximate within any factor when the structure is a general graph, and the diffusion model is LTM. We also considered the problem when the structure is an arborescence, and the diffusion process follows the ICM rules. We showed that the problem is inapproximable within any factor, except $P = NP$. Another structure that we investigated is a tree where the voting system is a variation of straight-party voting. We presented a polynomial-time algorithm to maximize the expected score of our target party regarding both LT and IC diffusion models. It yields that we can get a $\frac{1}{3}$-approximation factor for maximizing MoV in constructive case, and $\frac{1}{2}$-approximation factor concerning MoV in the destructive model.

The results of this paper open several research directions. Considering the multi-winner election control through social influence on arborescence, when the diffusion model is LTM can be an exciting research problem. We conjecture that maximizing both objective functions (MoV and DoW) is hard; however, there exists a polynomial-time algorithm for the IM problem on arborescence under LTM. We plan to consider maximizing MoV in straight-party voting to either present an optimal solution or provide a hardness result regarding both constructive and destructive cases. Furthermore, maximizing DoV on the bidirected trees, where a child can activate its parent too, can be impressive.

We conjecture that the problem accepts a polynomial-time algorithm following a similar dynamic programming approach.

**Author Contributions:** Conceptualization, M.A.M. and G.D.; methodology, M.A.M. and G.D.; software, M.A.M. and G.D.; validation, M.A.M. and G.D.; formal analysis, M.A.M. and G.D.; investigation, M.A.M. and G.D.; resources, M.A.M. and G.D.; data curation, M.A.M. and G.D.; writing–original draft preparation, M.A.M. and G.D.; writing–review and editing, M.A.M. and G.D.; visualization, M.A.M and G.D.; supervision, G.D.; project administration, G.D.; funding acquisition, G.D. All authors have read and agreed to the published version of the manuscript.

**Funding:** This work has been partially supported by the Italian MIUR PRIN 2017 Project ALGADIMAR "Algorithms, Games, and Digital Markets".

**Conflicts of Interest:** The authors declare no conflict of interest.

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
