# Peer review of "Multi-Winner Election Control via Social Influence: Hardness and Algorithms for Restricted Cases"

_algorithms, doi:10.3390/a13100251_

Round 1

Reviewer 1 Report

In this paper, authors proposed multi-winner election control through SI on different graph structures and diffusion models, and our goal is to maximize/minimize the number of winners in our target party. They proved the problem is hard to approximate within any factor when the structure is a general graph, and the diffusion model is LTM. They also considered the problem when the structure is an arborescence, and the diffusion process follows the ICM rules. Finally, they conjecture that the problem accepts a polynomial-time algorithm following a similar dynamic programming approach.

This is a well-written paper and the results are interesting. The readers should find it easy to follow. The paper fits well in the lines of the readers of the Journal. It gives credit to previous work and its length is adequate. The title and abstract are adequate to the content of the paper. Overall I recommend publication after minor revision.

Minor Comments:

  1. Authors need to give some practical examples to explain their problems.

Author Response

Dear reviewer,

We appreciate your consideration of this work. Regarding your comment, we have added one example for each diffusion model (pages 5 and 6), making the problems clear.

Regards,

Reviewer 2 Report

This manuscript considers multi-winner election control through social influence by extending the Linear Threshold Model and the Independent Cascade Model. After describing the model, they define two types of problems (constructive margin of victory problem and destructive margin of victory problem) and they proved that those problems are NP-hard to approximate within any factor of approximation. They presented a dynamic programming algorithm for the cases that the voting system is a variation of straight-party voting, and voters form a tree.

I positively assess that this manuscript is well-written including a significant research topic, a sufficient review, clear research question, suitable models and its analysis, a new algorithm proposed, and well-streamed structure. Therefore, I approve the publication in Algorithms.

If possible, the authors shall consider the following minor points.
1) There are many abbreviations. The readers may lose their way.
2) A paragraph must consists of multiple sentences. See lines 87-88.

Author Response

Dear reviewer,

Many thanks for your consideration. Regarding your comments:

1) We have replaced many of the abbreviations with their original text (see the attached file).

2) Fixed.

Regards,

This manuscript is a resubmission of an earlier submission. The following is a list of the peer review reports and author responses from that submission.